

# Using snow depth observation to provide insight into the quality of regional-scale snowpack simulations for avalanche forecasting

Simon Horton[1,2] and Pascal Haegeli[1]

[1]Simon Fraser University, Burnaby, BC, Canada
[2]Avalanche Canada, Revelstoke, BC, Canada

**Correspondence:** Simon Horton (shorton@avalanche.ca)

**Abstract.** The combination of numerical weather prediction and snowpack models has potential to provide valuable information about snow avalanche conditions in remote areas. However, the output of snowpack models is sensitive to precipitation inputs, which can be difficult to verify in mountainous regions. To examine how existing observation networks can help interpret the accuracy of snowpack models, we compared snow depths predicted by a weather-snowpack model chain with data from automated weather stations and manual observations. Data from the 2020-21 winter were compiled for 21 avalanche forecast regions across western Canada covering a range of climates and observation networks. To perform regional-scale comparisons, snowpack model simulations were run at select grid points from the HRDPS numerical weather prediction model to represent conditions at treeline elevations and observed snow depths were interpolated to the same locations. Snow depths in the Coast Mountain range were systematically overpredicted, while snow depths in many parts of the interior Rocky Mountain range were underpredicted. The impact of these biases had a greater impact on the simulated avalanche conditions in the interior ranges, where faceting was more sensitive to snow depth. To put the comparisons in context, the quality of the observations were assessed with uncertainties in the interpolations and by checking whether snow depth increases during stormy periods were consistent with the forecast avalanche hazard. While some regions had high quality observations, many regions had large uncertainties, suggesting in some situations the modelled snow depths could be more reliable than the observations. The analysis provides insights into the potential for validating weather and snowpack models with readily available observations, and for how avalanche forecasters can better interpret the accuracy of snowpack simulations.

## 1 Introduction

Numerical weather prediction (NWP) models provide valuable information to avalanche forecasters, as avalanche conditions are heavily influenced by the evolution of weather patterns. With years of operational experience, forecasters develop a grounded understanding of how well specific NWP models predict weather in their local mountains. Predicting snowpack conditions in a similar way may be possible by forcing snowpack evolution models such as SNOWPACK or Crocus with weather data from NWP models (Morin et al., 2020). However, developing operational trust and understanding in snowpack models is difficult due to the complexity and spatial variability of the snowpack.



Efforts to verify snowpack models have faced challenges due to uncertainties with verification data sets and a lack of
objective verification frameworks (Morin et al., 2020). Approaches in academic literature have included verifying model
output with regional-scale avalanche hazard assessments (Bellaire et al., 2017; Giraud et al., 1987), satellite-retrieved optical
properties (Charrois et al., 2016; Cluzet et al., 2020), snow profile stratigraphy observations (Bellaire and Jamieson, 2013;
Brun et al., 1992; Durand et al., 1999; Lehning et al., 2002; Viallon-Galinier et al., 2020), snowpack stability tests (Reuter
et al., 2015), and with snow height and snow water equivalent observations (Bellaire et al., 2011; Durand et al., 2009; Lafaysse
et al., 2013; Schmucki et al., 2015; Winstral et al., 2018). These studies primarily represent case studies at specific locations
for specific periods when detailed verification data were collected. While these types of studies are valuable for assessing
the general skill of snowpack models, they offer limited operational insight about when snowpack models can be trusted for
avalanche forecasting. Producing such operational insight requires a real-time verification framework based on continuous data
streams.

Several types of observation data are available to avalanche forecasters, including manual observations of weather, snowpack,
and avalanches from field observers and continuous data streams from automated weather stations and avalanche detection
networks. While each of these data streams could provide verification data for snowpack models, all observation networks are
limited in their spatial-temporal coverage and their representativeness of regional-scale avalanche conditions. Lundquist et al.
(2020) identify similar challenges with observation networks for related applications such as mountain hydrology and ecology,
and therefore conclude that the skill in modelling mountain precipitation is now comparable or superior to observation networks
in many contexts. Accordingly, in some contexts, it could be misleading to assume field observations are more representative
than model simulations.

Sensitivity studies consistently find precipitation input as the main driver of uncertainty in snowpack models (Raleigh et al.,
2015; Richter et al., 2020). Observations of winter precipitation are available from different types of measurements including
cumulative precipitation from rain gauges, snow water equivalent from snow pillows, and snow depth from acoustic sensors or
manual probing (Wang et al., 2017). Of these, snow depth observations are typically the most abundant and representative type
of precipitation observation available in most avalanche forecasting regions, and accordingly could be a relatively simple data
stream to perform operational snowpack model verification.

In this study, snow depth observations were compiled from existing networks used by avalanche forecasters across western
Canada to assess the accuracy of snowpack models forced with NWP inputs. The objective was to investigate the potential of
verifying snowpack models in real-time with snow depth observations. This included assessing the reliability of the observations,
comparing modelled and observed snow depths across space and time at a regional-scale, and investigating the impacts of
incorrect snow depths on the resulting snowpack stratigraphy. The results provide insights into how avalanche forecasters can
better interpret the accuracy of snowpack simulations and highlight the potential for snow depth observations to verify and
improve NWP and snowpack models in mountainous terrain.



## 2   Data

The three key datasets for this study were snowpack simulations, snow depth observations, and avalanche hazard assessments. This section outlines how each of these datasets were compiled to evaluate snowpack simulations across western Canada for the 2020-21 winter season. The study period was restricted to four winter months when regular manual observations and hazard assessments were available (2020-12-01 to 2021-03-31).

### 2.1   Study area

Data were compiled for 21 public avalanche forecast regions in British Columbia, Alberta, and Yukon (Fig. 1a). The regions were grouped into three mountain ranges based on their predominant snow climate characteristics – the maritime Coast Mountains, the transitional Columbia Mountains, and the continental Rocky Mountains (Shandro and Haegeli, 2018). To collect data that was representative of avalanche conditions in these regions, we focused the analysis on conditions at treeline elevations. Avalanche forecasters classify terrain into alpine, treeline, and below treeline bands based on terrain and vegetation characteristics. The snowpack at treeline tends to be sheltered from the wind and therefore more homogenous than in the alpine. Since there are no strict definitions for the boundaries of the vegetation bands, and there is some subjectivity in how they are used operationally, we had to derive an objective reference for the treeline elevation for the purpose of this study. Publicly available digital elevation model and land cover classification data were used to search for the upper extent of forested terrain throughout the study area. First, pixels with any type of forested land cover were extracted from the 30 m resolution 2015 Land Cover of North America (North American Land Change Monitoring System, 2015). The elevation of these pixels was extracted from a global digital elevation model (Danielson and Gesch, 2011) with 30 arc-second resolution (approximately 145 m). The treeline elevation at each pixel was derived by searching for the maximum elevation of forested terrain within a 10 x 10 km moving window across the study. The resulting elevations ranged from 1100 m in the northwest to 2200 m in the southeast (Fig. 1b). Feedback from avalanche forecasters suggested this method produced elevations that roughly aligned with their understanding of vegetation bands in western Canada.

### 2.2   Snowpack simulations

Snowpack simulations were produced by forcing SNOWPACK (Lehning et al., 1999) with output from the High-Resolution Deterministic Prediction System (HRDPS), an operational NWP model run by the Canadian Meteorological Centre on a 2.5 km horizontal grid (Milbrandt et al., 2016). Simulations were configured to represent treeline snowpack conditions across the 21 avalanche forecast regions.

Rather than running simulations at all NWP grid points within the regions, 856 grid points were sampled to represent conditions at treeline (Fig. 1b). The forecast regions were split on a 10 x 10 km grid and then the average treeline elevation was computed in each grid cell. The HRDPS grid point with the closest match to the computed treeline elevation was selected for each grid cell. Grid cells were discarded if no grid points were within 100 vertical metres of the treeline elevation, which occurred 24% of the time (either because the NWP model topography was smoother than the real terrain or because the grid cell



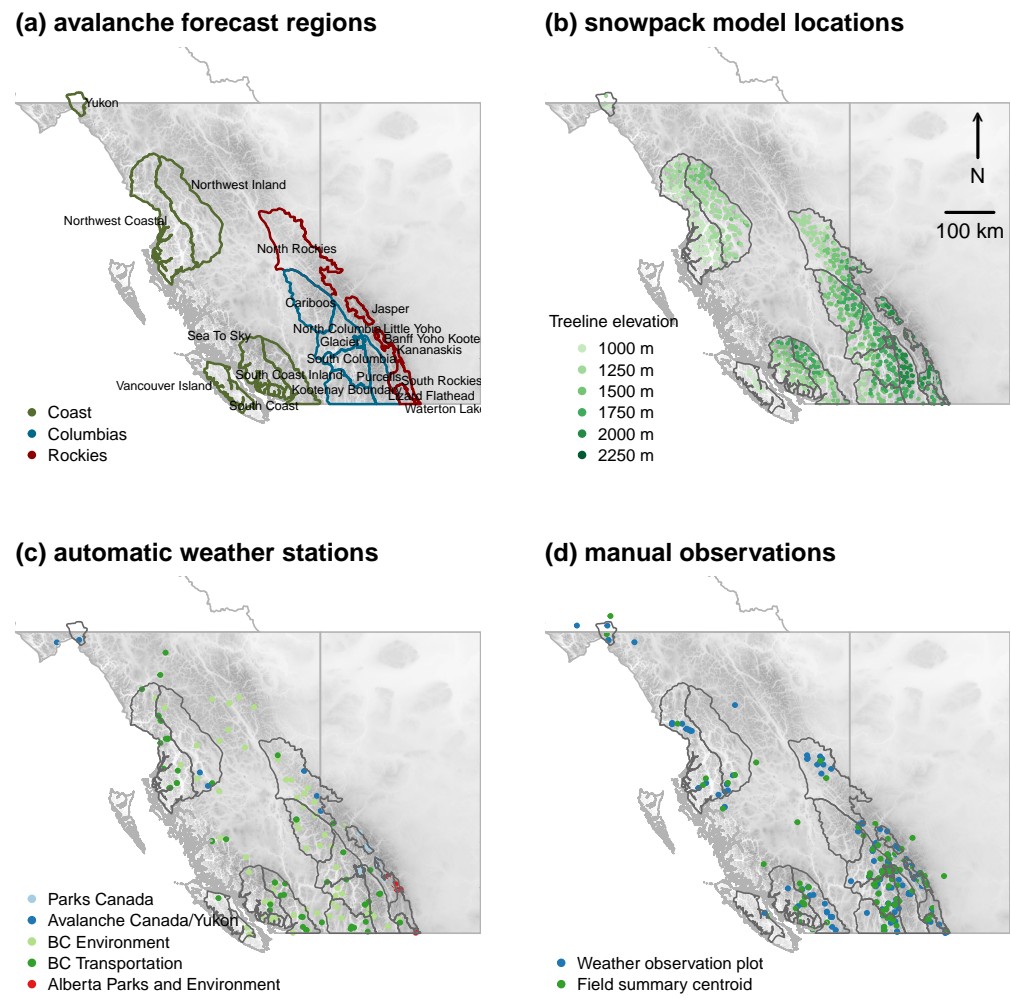

**Figure 1.** Location of (a) avalanche forecast regions, (b) snowpack model simulation grid points colour-coded by treeline elevation at that location, (c) automatic weather stations, and (d) manual observations and field summaries.



did not actually contain high elevation terrain). Selecting a single grid point in each 10 x 10 km cell was found to effectively reduce redundant information, as there were minimal variations between HRPDS grid points at similar elevations within the same grid cell.

Snowpack simulations were produced by concatenating forecasts from each operational run of the HRDPS, starting on 1 September 2020. The HRDPS is initialized every six hours, and SNOWPACK was forced with data from the 6 to 12 predictive hours from each run. SNOWPACK version 3.6 was configured with default settings for flat-field profiles and wind transport disabled. The height of snowpack (HS), hereafter referred to as snow depth, and the height of new snow over 24 h periods (HN) were output for each day of the study period at 0 UTC (16:00/17:00 local time).

### 2.3 Snow depth observations

Snow depth observations were compiled from three sources: automated weather stations, manual observations, and field summaries. The following sections describe how each of these sources were compiled to obtain daily values of snow depth (HS) and height of new snow (HN) for each day of the study period at 0 UTC.

### 2.3.1 Automated weather stations

Weather data was queried from an Avalanche Canada database that aggregates automated weather stations (AWS) deemed relevant for operational avalanche forecasting (Fig. 1c). 146 of the 238 AWS in this database are equipped with acoustic snow depth sensors. These stations are operated by various agencies including Parks Canada (14 stations), British Columbia Ministry of Transportation and Infrastructure (63 stations), British Columbia Ministry of Environment (56 stations), Alberta Parks and Environment (6 stations), Avalanche Canada (5 stations), and the Yukon Avalanche Association (2 stations). While some networks already apply quality control to the data, an additional spike removal filter was applied to remove observations where HS increased or decreased more than 10 cm in one hour. HS recordings were extracted at 0 UTC each day. Calculating height of new snow (HN) from AWS data is challenging due to settlement and instrument noise (Wang et al., 2017). For this study, HN was calculated by summing hourly HS changes over 24 h periods. HN was set to zero when this sum was negative. Summing both positive and negative changes was found to be more reliable than summing only positive changes, as instrument noise frequently resulted in HS increases that would appear as several centimetres of new snow on days when precipitation did not occur. The disadvantage of this method is that HN is underestimated on days with significant snowpack settlement. The number of daily observations fluctuated over the study period due to sensor and transmission errors, resulting in a median of 129 HS observations and 106 HN observations per day.

### 2.3.2 Manual observations

Avalanche safety operations report manual weather observations on the Canadian Avalanche Association's Information Exchange (InfoEx). Weather observations consist of manual measurements taken at fixed instrumented study plots following standards published by the Canadian Avalanche Association (2016). HS is measured on a permanent stake and HN is measured with a





ruler on a snow board that is cleared daily. While observations are typically made twice per day, only afternoon observations

were included in this study to be consistent with the timing of the other data sources. Over the entire study period, 94 operations reported observations from 197 different study plots with known geographic coordinates and elevations (Fig. 1d). The number of daily observations fluctuated over the study period, with a median of 75 HS observations and 58 HN observations per day.

### 2.3.3 Field summaries

Avalanche operations also report field weather summaries to InfoEx, which are distinct from weather observations made at

125 fixed study plots. Field weather summaries summarize the range of conditions encountered in a broad geographic area while travelling in the mountains (Canadian Avalanche Association, 2016). Field summaries include subjective and spatially broader estimates of HS and HN that are relevant for assessing avalanche hazard in that area, often based on several measurements made throughout the day. The geographic extent of these observations is less precise than manual observations and are defined by polygons covering their operating areas and with elevation ranges that typically span treeline elevation. Fig. 1d shows the

130 polygon centroids for 99 operations that reported field summaries over the study period. Fewer operations submitted field summaries than manual observations, with a median of 19 HS observations and 16 HN observations per day.

### 2.4 Avalanche hazard assessments

Avalanche hazard assessments were compiled to assess how well modelled and observed snow data captured impactful snowfall events. Daily forecasts were compiled for all 21 forecast regions over the study period. Forecasts are published following the

135 workflow described by the conceptual model of avalanche hazard (Statham et al., 2018), and include a nowcast assessment of current danger ratings and avalanche problems. To quantify the impact of snowfall events, days with storm slab avalanche problems at treeline were identified. Storm slab problems have the most direct link with new snow under the North American avalanche problem definitions (Statham et al., 2018), and Horton et al. (2020b) show a strong relationship between storm slab problems and height of new snow. Two components from the hazard assessments were extracted to characterize the impacts of

140 these snowfall events: the presence of a storm slab avalanche problem at treeline (a binary value of absent or present), and the danger rating at treeline on days with storm slab problems (ordinal values of 1-Low, 2-Moderate, 3-Considerable, 4-High, and 5-Extreme). The number of days with storm slab problems over the study period ranged from six for the Kananaskis region to 73 for the Northwest Coastal region.

## 3 Methods

This section describes how the datasets were prepared and analyzed to answer the following three questions:

1. How reliable were the observations?
2. How did modelled and observed snow depths compare across space and time?
3. How do incorrect snow depths impact the modelled snowpack stratigraphy?



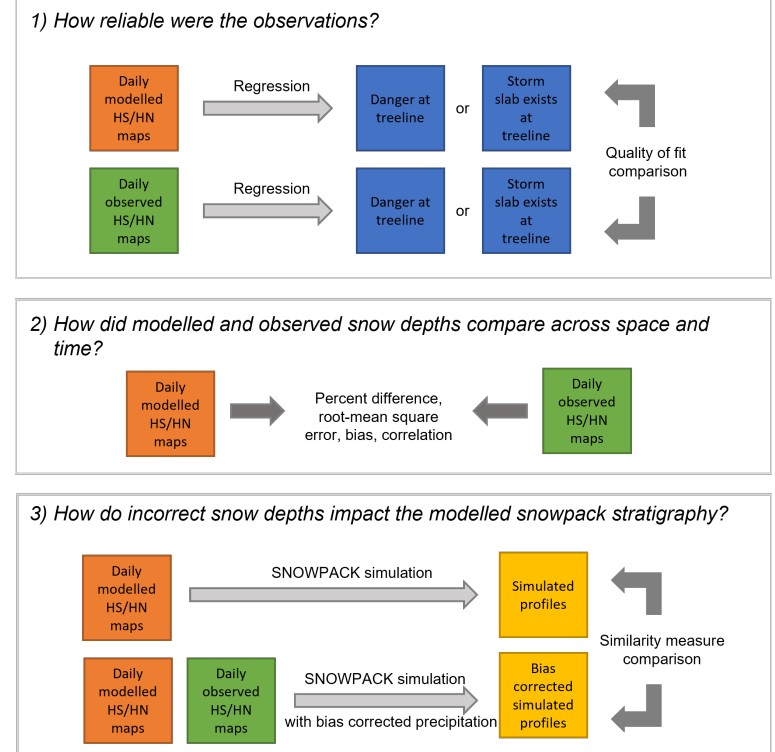

**Figure 2.** Methods used to analyze the reliability of observations, compare modelled and observed snow depths, and measure the impact of incorrect snow depths on snowpack simulations.

Fig. 2 summarizes the methods used to answer these questions. First, the observed snow depths were interpolated onto the
same grid as the snowpack models to produce daily maps of modelled and observed HS and HN to allow consistent analysis
across space and time (Sect. 3.1). The reliability of the observations was assessed with two methods; first, from the uncertainty
in the interpolations, and second, by fitting regression models to see whether avalanche conditions were better explained by
height of new snow from the modelled or observed data (Sect. 3.2). Basic statistical measures were used to compare the snow
depths across space and time (Sect. 3.3). Finally, to investigate the impact on snowpack stratigraphy, two sets of simulated
snow profiles were compared; one with the original NWP model inputs and the second with bias-corrected precipitation inputs
(Sect. 3.4).

### 3.1 Interpolating observations

Observed snow depths were aggregated and interpolated to the same spatial grid as the modelled data. The interpolations were
produced in two steps: first a lapse rate adjustment was applied to estimate the observed value at the local treeline elevation, then
a spatial interpolation was applied to predict snow depths at treeline elevations across the entire study area. The interpolation





was not designed to capture fine-scale patterns across terrain, but rather aggregate available observations into a consistent format for regional-scale comparisons.

Elevation corrections were applied to the AWS and manual observations to account for differences between the observation elevation ($z_{obs}$) and the treeline elevation at that location ($z_{treeline}$). Field summaries were not adjusted as their elevations
were unknown, but are intended to represent conditions at treeline elevations (Canadian Avalanche Association, 2016). HS was lapse rate adjusted with the exponential precipitation adjustment factor proposed by Thornton et al. (1997) which has been commonly applied in hydrological models (e.g., Liston and Elder, 2006; Schirmer and Jamieson, 2015):

$$HS_{treeline} = HS_{obs} \frac{(1+0.35(z_{treeline}-z_{obs}))}{(1-0.35(z_{treeline}-z_{obs}))} \tag{1}$$

where $HS_{treeline}$ is the adjusted snow depth at treeline derived from an original observed depth $HS_{treeline}$. A lapse rate factor
of 0.35 was suggested for winter months in western USA by Thornton et al. (1997). Since only observations within 500 vertical metres of treeline were included, snow depths were multiplied by a factor of 0.70 to 1.42. For the entire set of observations, the average snow depth increased by a factor of 1.18 after applying treeline adjustments. The same lapse rate adjustment was applied to height of new snow where $HS$ in Eq. 1 was replaced with $HN$.

Spatial interpolations were applied to the treeline corrected HS and HN observations with simple kriging. The gstat package
for R was used to fit a unique variogram model for HS and HN on each day of the study period to describe the structure of the spatial correlation (Pebesma, 2004). Given the sparsity of observations, the best fitting of three possible variogram models were chosen (spherical, exponential, and pure nugget models) where the best fit was chosen according gstat's default method of weighting residuals (i.e., number of pairs in a bin divided by the square of the bin's distance). Simple kriging was applied with the best variogram model to predict HS and HN at all 856 model grid points for each day of the study period. Each prediction
combined all available observations from AWS, manual observations, and field summaries.

The kriging predictions also provided an estimate of prediction variance ($\sigma^2_{HS}$), which was used to estimate the uncertainty in observed snow depth. The relative kriging standard deviation ($RKSD$) was defined at the square root of the prediction variance divided by the predicted height:

$$RKSD = \frac{\sqrt{\sigma^2_{HS}}}{HS} \tag{2}$$

The $RKSD$ is a relative measure of uncertainty where small values suggest the interpolation error was small relative to the predicted snow depth and large values suggest the error was large relative to the predicted snow depth.

### 3.2 Regression models relating height of new snow to avalanche conditions

To assess the reliability of the modelled and observed data sets, regression models were fit to predict the regional-scale avalanche conditions from height of new snow. Height of new snow has been shown to be a strong predictor of avalanche
release, as numerous studies have highlighted the link between new snow with avalanche danger (Schirmer et al., 2010) and the presence of storm slab avalanche problems (Horton et al., 2020b). Therefore, for this analysis we assume that increases in snow depth should be proportional to the likelihood of storm slab avalanche problems and the likelihood of increased avalanche danger.





To test this assumption, regression models were fit to predict (1) the presence of storm slab problems, and (2) the danger

rating using height of new snow over three days (HN(3d)) as the single predictor. Height of new snow over three days was

calculated by adding the current day's HN with the HN from the previous two days. For each forecast region, a logistic

regression model with a single predictor was fit to predict the probability of a storm slab avalanche problem being present

based on the modelled HN(3d) at grid points in the region and an analogous logistic regression model was fit using the observed

HN(3d) at the same grid points. The goodness of fit of the two models were compared with McFadden's pseudo R-squared

measure (an analogous value to the coefficient of determination that can be applied to general linear models) to assess whether

the modelled or observed HN(3d) explained storm slab presence better in each forecast region.

A similar approach was used to fit ordinal regression models in each forecast region to predict the probability of danger

rating increasing with HN(3d). These models were restricted to days when storm slab avalanche problems were present, as the

danger would more likely be driven by other factors on days without storm slab problems. The ordinal regression models were

205 fit using the cumulative link model from the ordinal package for R (Christensen, 2019). Again, McFadden's pseudo R-squared

measure was used to determine which dataset explained danger ratings better in each region.

### 3.3 Comparing snow depths

The continuous grids of modelled and observed snow depths allowed plotting spatial comparison on maps and temporal

comparisons on timeseries. Modelled and observed snow depths were compared with basic quantitative metrics, including

210 the percent difference to provide a relative measure of the differences, root mean square error to assess the magnitude of

the differences, bias to assess the prevailing direction of the differences, and Spearman rank-order correlation coefficient to

determine whether a set of modelled and observed snow depths increased at the same locations or times. These statistics are

defined as follows:

$$\text{Percent difference (\%)} = \frac{(HS_{mod} - HS_{obs})}{HS_{obs}} * 100\% \tag{3}$$

$$\text{Root mean square error (m)} = \sqrt{\frac{1}{n} \sum (HS_{mod} - HS_{obs})^2} \tag{4}$$

$$\text{Bias (m)} = \frac{1}{n} \sum (HS_{mod} - HS_{obs}) \tag{5}$$

where $HS_{mod}$ is the modelled snow depth, $HS_{obs}$ is the observed snow depth, and $n$ is the number of observations. The

Spearman rank-ordered correlation coefficient is defined as the Pearson correlation between the rank values of $HS_{mod}$ and

$HS_{obs}$, rather than the correlation between the values themselves. The same statistics were calculated for height of new snow

$HN$.

### 3.4 Correcting simulated snow profiles

To illustrate the impact incorrect snow depths could have on the simulated snowpack structure, a sample of locations were

chosen to test a simple bias correction method. One representative profile location was chosen for each region by selecting the

grid point that most frequently had the median modelled snow depth over the study period. A bias correction factor ($k$) was





calculated for each location according to

$$k = \overline{\frac{HS_{obs}}{HS_{mod}}} \tag{6}$$

based on the modelled and observed snow depths. Averaging the factor over each day of the study period smoothed out variations that appeared on individual days. The hourly precipitation inputs at each grid point location were multiplied by this correction factor, and then SNOWPACK was re-run with the bias-corrected precipitation.

The original and bias-corrected profile simulations were compared by applying the similarity measure introduced by Herla et al. (2021). The similarity measure performs a pairwise comparison of two profiles from a perspective of stratigraphy features relevant to avalanche hazard. To calculate the similarity measure, the algorithm first aligns each layer with dynamic time warping, where the deposition date, grain type, and hardness are used to match layers in the two profiles. After alignment, the algorithm compares the similarity of the grain type and hardness of aligned layers, while putting more weight on weak layers,

melt-freeze crusts, and new snow layers. The similarity measure values range from 0 to 1, where 0 corresponds to highly dissimilar profiles and 1 corresponds to identical profiles. The similarity value of the original and bias-corrected profiles were computed for each day of the study period to quantify the impact adjusting precipitation inputs had on the interpretation of avalanche hazard conditions.

## 4    Results

### 4.1    Reliability of snow depth observations

Snow depths were interpolated with greater confidence in areas with dense observation networks (Fig. 3), such as the southern Coast range and the central Columbias. In these areas, the average standard deviation of interpolated snow depths was roughly 30-40% of the snow depth, while areas with the greatest uncertainty had average RKSD values up to 50-60%. Areas with low confidence in interpolated snow depth included northern parts within each coastal region and the southern regions of the

245 Columbias and Rockies. The median RKSD for all 856 grid points over the study period was 46%, suggesting the aggregation of snow depths from available observations had considerable uncertainty across many forecast regions.

Predicting avalanche conditions with height of new snow over three days yielded comparable results with modelled and observed data. Logistic regression models predicting the presence or absence of storm slab avalanche problems performed better with modelled data in 10 of 21 regions (Table 1). This included most coastal regions and several of the southern interior

regions. Ordinal regression models predicting danger ratings performed better with modelled data in 11 of 21 regions. This included the same coastal regions, fewer of the southern interior regions, and the addition of some of the northern interior regions.

In the interior, the regions where avalanche conditions were predicted better with observation data were typically the regions relatively smaller RKSD (e.g., Glacier and North Columbia), suggesting the weather observation networks are relatively

representative of avalanche conditions in these regions. However, the southern coastal regions also had areas with relatively





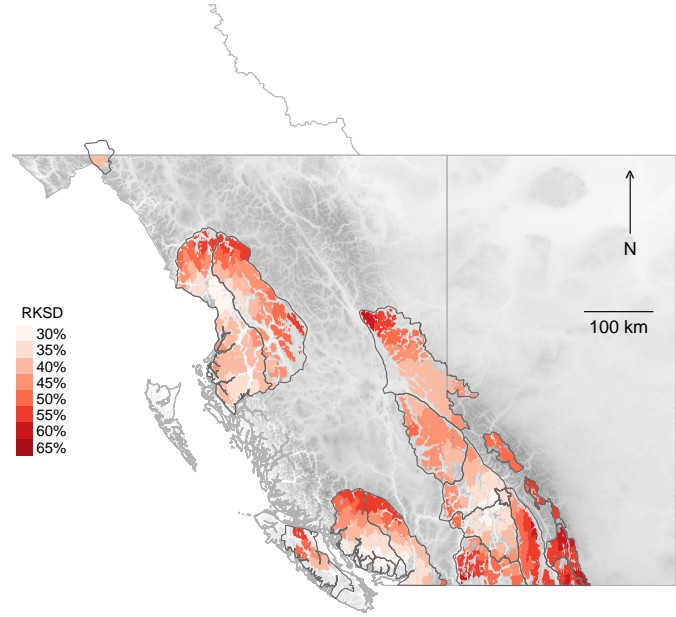

**Figure 3.** The average relative kriging standard deviation over the study period shows areas where there was relatively high or low confidence in the snow depths interpolated from observations.

small RKSD, yet avalanche conditions were predicted better with modelled snowfall, suggesting that even though the weather observation networks were relatively consistent they may be less representative of conditions in avalanche terrain.

## 4.2 Comparing modelled and observed snow depths

### 4.2.1 Regional-scale spatial patterns

Snow depths on 31 March 2021 are mapped in Fig. 4 to illustrate regional-scale patterns by the end of the winter season. Observed snow depths were relatively deep (i.e., greater than 300 cm) along western parts of the Coast range as well as the central parts of the Columbias. Shallower snow depths were observed on the eastern side of the Coast range, the perimeters of the Columbias, and throughout most of the Rockies (Fig. 4a-b). Modelled snow depths had similar regional-scale patterns, but with more extreme differences between the deep and shallow snowpack areas within each region (Fig. 4c). For example, 265 observed snow depths ranged from 115 to 491 cm across the entire study area, while modelled snow depths ranged from 39 to 917 cm. Despite the different ranges, the median snow depths were similar with values of 257 and 285 cm for the observed and modelled data, respectively.





**(a) observed snow depth**

**(b) snow depth interpolated from observations**

**(c) modelled snow depth**

**(d) percent difference in snow depth (model – obs)**

**Figure 4.** Comparison of observed and modelled snow depths on 31 March 2021 with (a) original snow depths from 166 observations, (b) the observed snow depths interpolated to treeline elevations at 856 NWP model grid point locations, (c) the modelled snow depth at the same grid points, and (d) the percent difference between the modelled and observed snow depths (positive values indiciate modelled depths are greater than observed).



**Table 1.** Comparison of regression models predicting the presence of storm slab avalanche problems and danger ratings with modelled versus observed snow data. McFadden's pseudo R-squared values are listed for each model to compare their goodness of fit with asterisks identifying the model with a better fit.

| Range | Region | Storm slab problem | | Danger rating | |
| --- | --- | --- | --- | --- | --- |
| | | Observed | Modelled | Observed | Modelled |
| Coast | Yukon | 0.05 | 0.2* | 0 | 0.03* |
| | Northwest Coastal | 0.29 | 0.32* | 0.02 | 0.04* |
| | Northwest Inland | 0.24* | 0.08 | 0.04* | 0.04 |
| | Vancouver Island | 0.03 | 0.07* | 0.04 | 0.08* |
| | South Coast | 0.28 | 0.32* | 0.01 | 0.07* |
| | Sea To Sky | 0.27 | 0.37* | 0.04 | 0.07* |
| | South Coast Inland | 0.32* | 0.24 | 0.06* | 0.06 |
| Columbias | Cariboos | 0.36* | 0.29 | 0.12 | 0.17* |
| | North Columbia | 0.38* | 0.34 | 0.2* | 0.15 |
| | Glacier | 0.4* | 0.32 | 0.21* | 0.14 |
| | South Columbia | 0.31 | 0.32* | 0.2* | 0.12 |
| | Kootenay Boundary | 0.38* | 0.32 | 0.13* | 0.05 |
| | Purcells | 0.22 | 0.23* | 0.27* | 0.18 |
| Rockies | North Rockies | 0.26* | 0.21 | 0.06 | 0.06* |
| | Jasper | 0.15* | 0.1 | 0.06* | 0.05 |
| | Banff Yoho Kootenay | 0.17* | 0.14 | 0.2 | 0.23* |
| | Little Yoho | 0.26 | 0.29* | 0.21 | 0.26* |
| | Kananaskis | 0.17* | 0.14 | 0.31* | 0.3 |
| | Lizard Flathead | 0.35 | 0.37* | 0.04 | 0.09* |
| | South Rockies | 0.32 | 0.36* | 0.03 | 0.08* |
| | Waterton Lakes | 0.15* | 0.07 | 0.22* | 0.09 |

The percent difference between modelled and observed snow depths show where regional-scale discrepancies were most pronounced (Fig. 4d). Modelled snow depths were substantially greater than local observations in the Coast range, especially on the western (upslope) side of the range. With large snow depths in the Coast range, the root mean square errors were relatively large and exceeded 100 cm in many regions (Table 2). Snow depth biases were positive in all Coast regions, except for the Yukon. Correlations between modelled and observed snow depths within each Coast region ranged from 0.25 to 0.75, suggesting there was moderate to strong agreement on the location of the relatively deeper and shallower areas within each region.



**Table 2.** Statistics comparing modelled and observed snow depths within each forecast region on 31 March 2021. (Footnote: [a] Number of observations within forecast region boundaries prior to interpolation. The interpolated observations also consider observations from neighbouring areas.)

| Range | Region | Observations[a] | Number of grid points | Root mean square error (cm) | Bias (cm) | Correlation |
|-------|--------|------------------|------------------------|------------------------------|-----------|-------------|
| Coast | Yukon | 1 | 6 | 113 | -74 | 0.75 |
| | Northwest Coastal | 18 | 149 | 205 | 127 | 0.40 |
| | Vancouver Island | 2 | 17 | 101 | 9 | 0.25 |
| | South Coast | 5 | 6 | 66 | 2 | 0.66 |
| | Sea To Sky | 6 | 62 | 194 | 131 | 0.33 |
| | South Coast Inland | 10 | 62 | 88 | -1 | 0.59 |
| Columbias | Cariboos | 10 | 72 | 79 | 21 | 0.08 |
| | North Columbia | 8 | 52 | 74 | -15 | 0.59 |
| | Glacier | 7 | 3 | 19 | -4 | 0.50 |
| | South Columbia | 18 | 50 | 81 | -10 | 0.46 |
| | Kootenay Boundary | 13 | 43 | 70 | -51 | 0.54 |
| | Purcells | 13 | 45 | 97 | 30 | 0.45 |
| Rockies | North Rockies | 14 | 104 | 91 | -32 | 0.42 |
| | Jasper | 9 | 13 | 56 | -14 | 0.45 |
| | Banff Yoho Kootenay | 11 | 3 | 29 | -9 | 0.50 |
| | Little Yoho | 0 | 2 | 48 | 46 | -1.00 |
| | Kananaskis | 5 | 2 | 147 | -144 | -1.00 |
| | Lizard Flathead | 10 | 10 | 33 | -23 | -0.15 |
| | South Rockies | 23 | 23 | 66 | -53 | -0.55 |
| | Waterton Lakes | 1 | 1 | 95 | -95 | NA |

In the Columbias, modelled and observed snow depths were relatively similar (Fig. 4d), although the modelled depths were substantially lower in the southwestern parts of the range and substantially higher in the western Purcell and central Cariboo regions. The root mean square error in Columbias ranged from 19 to 97 cm (Table 2), with relatively smaller biases than the other ranges. Positive correlations between 0.45 and 0.59 suggest the model and observation data had moderate agreement in where the relatively deeper and shallower snowpacks existed, except for the Cariboo region where the correlation was only

0.08.

In the Rockies, there was reasonable agreement between snow depths in the central regions of Banff Yoho Kootenay and Jasper, but modelled depths were substantially lower than observed in the southern and northern parts of the range (Fig. 4d). Negative biases existed in all regions except Little Yoho (Table 2), highlighting how modelled depths were systematically less

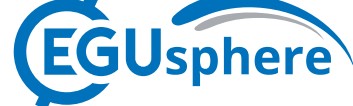

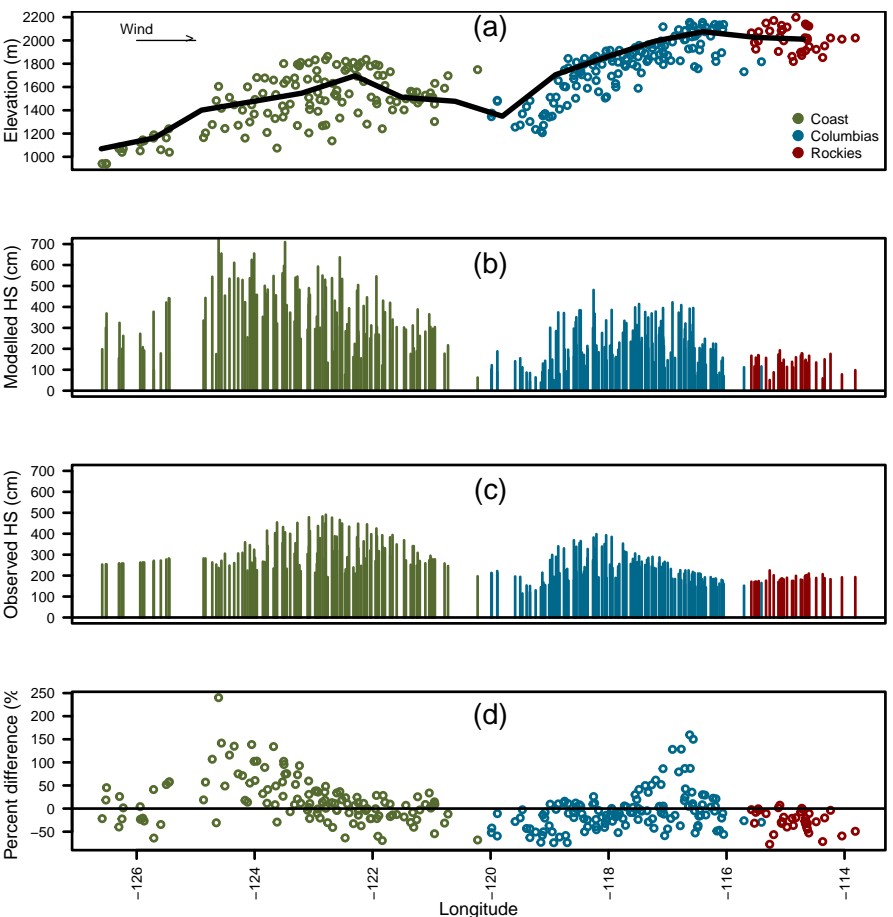

**Figure 5.** Latitudinal cross-section between 49 and 51 °N of (a) the elevation of NWP grid points, (b) modelled snow depth at treeline grid points, (c) observed snow depths interpolated to the same grid points, and (d) percent difference between modelled and observed snow depths.

than observed. While the regions in the southern Rockies had fewer grid points to compute correlations, their negative values
mean the model and observation data disagreed on where the relatively deeper and shallower snowpacks existed.

     To further illustrate the predominant spatial patterns across the three major mountain ranges, snow depths are plotted along
a latitudinal cross-section between 49 and 51 °N in Fig. 5. The western (windward) side of the Coast range had substantially
larger modelled snow depths than observed, while there was reasonable agreement along the highest terrain and the leeward
side of the Coast range. In the Columbias, the windward side had the greatest observed snow depths, which were slightly
larger than modelled, however, modelled depths were greater than observed over the highest terrain in the eastern side of the
Columbias. In the Rockies, there was minimal variation between the windward and leeward sides for both the modelled and
observed snow depths.



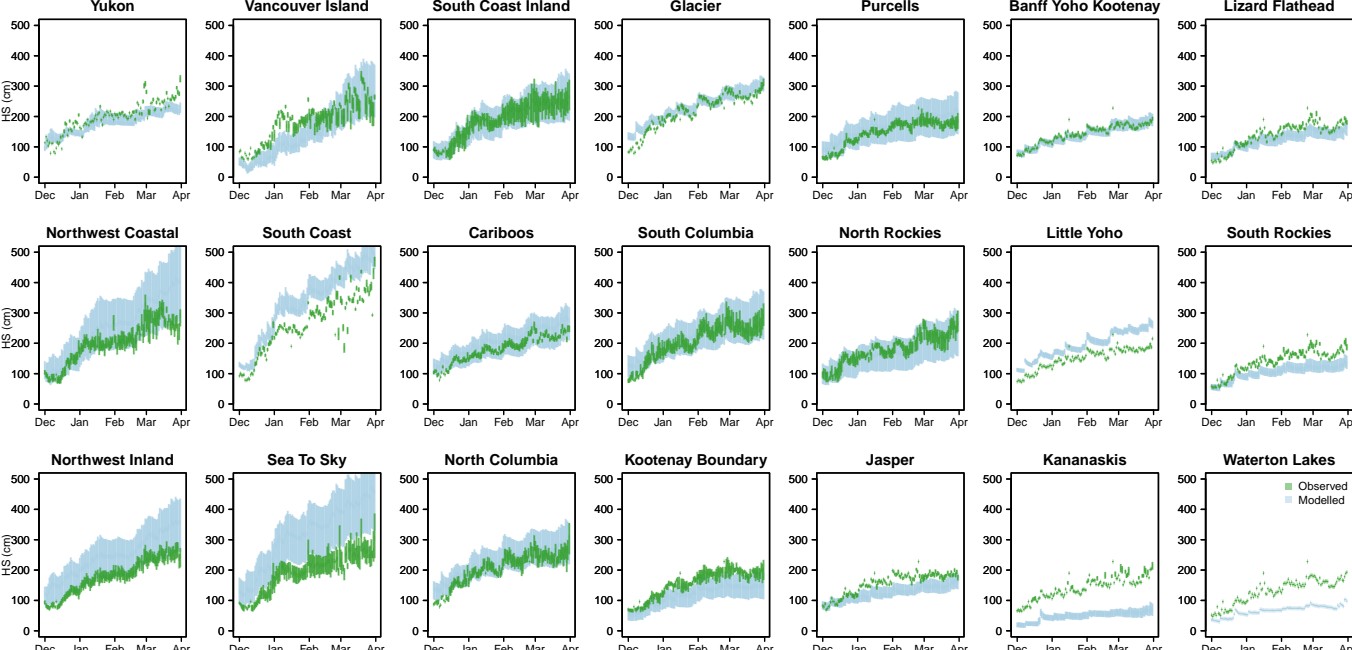

**Figure 6.** Interquartile range of snow depths for each avalanche forecast region throughout the study period from modelled (blue) and observed (green) data.

### 4.2.2 Temporal patterns

Timeseries of snow depths show the agreement between modelled and observed data over the course of the 2020-21 winter
(Fig. 6). Some regions such as the Cariboos, Glacier, and Banff Yoho Kootenay had relatively consistent agreement over
the season, while other regions, such as Sea to Sky and Waterton Lakes, had major discrepancies. Discrepancies arose from
both systematic biases that persisted for the duration of the season and from differences originating from specific events. For
example, systematic biases are evident in the Sea to Sky region where modelled depths were consistently greater than observed
and in the Kananaskis region where modelled depths were consistently less than observed. Examples of specific events causing
discrepancies include Vancouver Island where the observed depths increased in early January without a corresponding increase
in modelled depths, and in the Northwest Coastal region where changes in modelled and observed depths occurred at different
times throughout the season. The interquartile ranges of modelled snow depths were wider than the observations, especially in
regions with fewer observations.

Agreement in the timing of new snow was measured with the correlation between modelled and observed height of new
snow (HN) at each location (Fig. 7a). Across all 856 grid points, the median correlation was 0.69 and ranged from 0.26 to 0.91.
Correlations were strongest in the southern Coast range, the central Columbias, and the central Rockies and relatively weaker





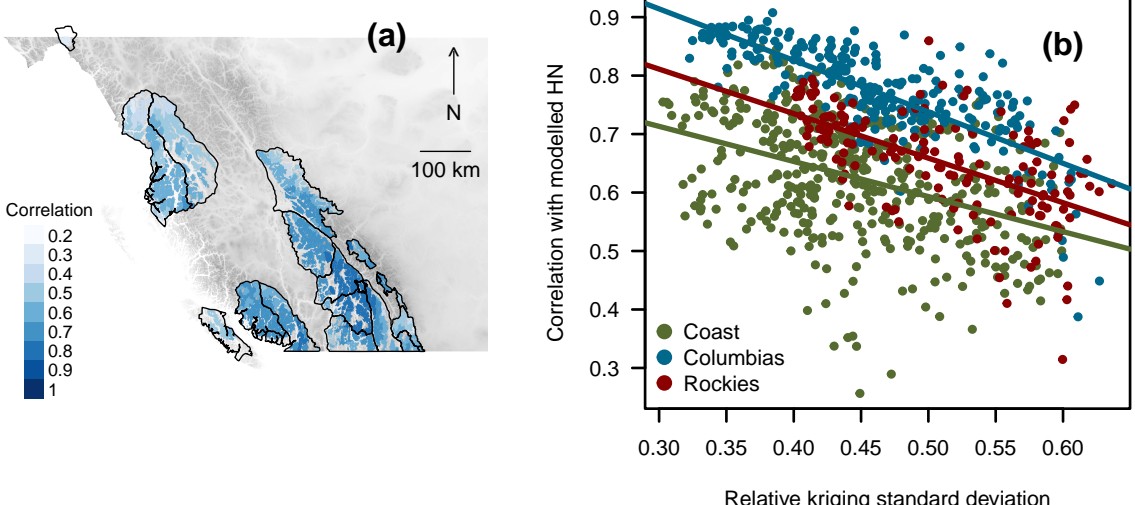

**Figure 7.** Correlation between modelled and observed height of new snow (HN) over the study period (a) mapped across the study area and (b) plotted against the uncertainty in the observations from the average relative kriging standard deviation (RKSD). Simple regression lines are fitted for each mountain range to show poorer HN correlations when observations had greater uncertainty.

in the northern Coast range and southern Rockies. The correlation was generally lower in areas with larger RKSD (Fig. 7b), suggesting disagreements were not only influenced by model errors but also by uncertainty in the observations.

## 4.3 Impact on modelled snowpack structure

### 4.3.1 Regional-scale snowpack patterns with NWP forcings

Snow profile simulations from each forecast region illustrate typical climatic patterns across western Canada (Fig. 8). The maritime climate in the Coast range resulted in thick layers of new snow, rounded grains, and melt forms. Profiles in the Columbias had transitional snowpacks with layers of rounded grains interspersed with thin layers of persistent grain types. The continental Rockies had the thinnest snowpacks and were largely composed of faceted grain types. Grain types in Fig. 8 are 315 colour-coded following the suggestions of Horton et al. (2020a) to highlight features relevant to avalanche conditions.

Plotting the snow depths interpolated from available observations at each profile location in Fig. 8 shows the potential accuracy of the simulations. The modelled stratigraphy was likely simulated better at locations where the observed and modelled snow depths closely agreed (e.g., South Coast Inland, Glacier, and Banff Yoho Kootenay regions). However, many locations had large discrepancies between modelled and observed snow depths, in which case it is less clear whether the model 320 was erroneous or the observations were not representative for that location.

The bias-correction factors calculated by comparing the modelled and observed depths at these locations with Eq. 6 ranged from 0.71 for the Sea to Sky location where observed depths were much smaller than modelled to 2.43 for the Kananaskis location where observed depths were much greater. The interquartile range of correction factors for these locations ranged



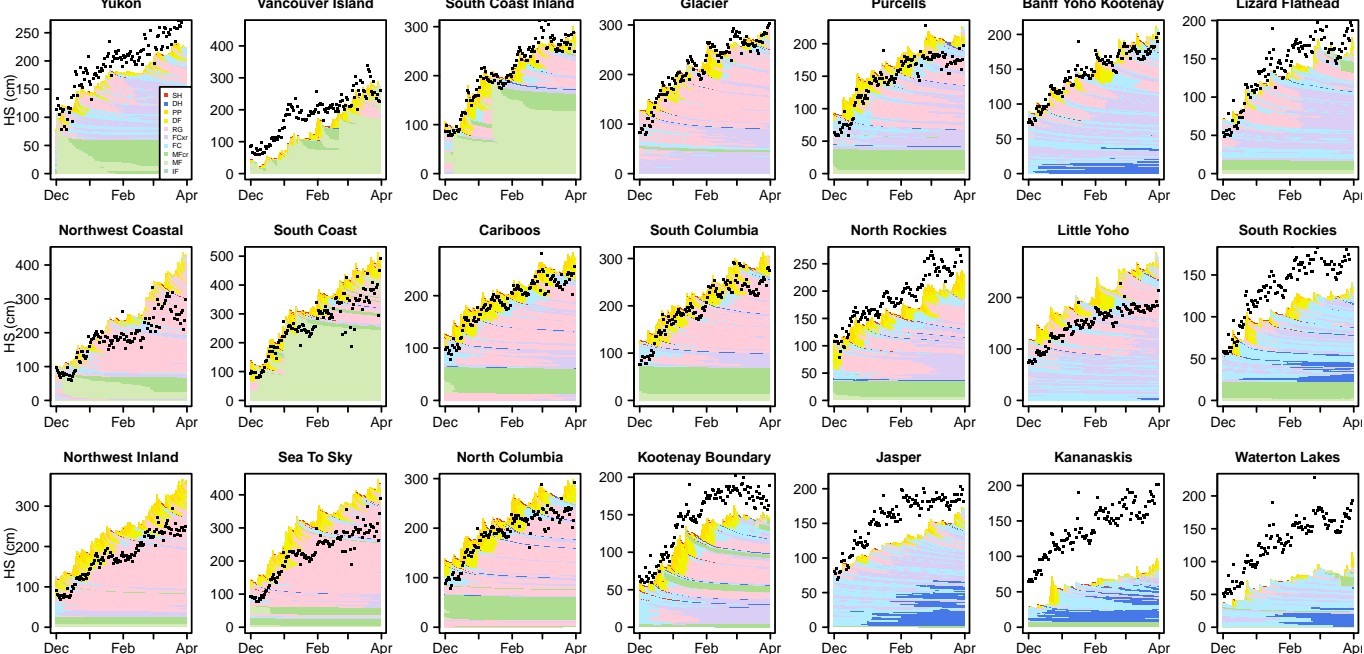

**Figure 8.** Simulated snowpack structure at a representative location in each forecast region. The snow depth interpolated from available observations at the same locations are shown with black squares to show the agreement between modelled and observed snow depths. Snowpack layers are colour-coded with surface hoar in red (SH), depth hoar in dark blue (DH), precipitation particles in bright yellow (PP), decomposing and fragmented precipitation particles in light yellow (DF), rounded grains in pink (RG), rounding faceted particles in mauve (FCxr), facet crystals in light blue (FC), melt-freeze crusts in bright green (MFcr), melt forms in light green (MF), and ice formations in blue-green (IF).

from 0.84 to 1.22. The interpolated snow depths at a single location were not consistent over time because the availability of
325 observations varied each day, hence averaging the bias-correction over the entire study period was deemed more appropriate than applying bias corrections at finer time scales.

### 4.3.2 Impact of bias-corrected precipitation inputs

Applying bias-corrections to the precipitation inputs resulted in a variety of impacts on the simulated snowpack structure. Profiles are compared for three specific cases in Fig. 9, including an example with a large reduction of precipitation for the
330 Sea to Sky profile ($k = 0.71$), a small reduction in precipitation for the Glacier profille ($k = 0.97$), and a large increase of precipitation for the Kananaskis profile ($k = 2.43$). Profiles for the remaining regions are shown in Appendix A.

The bias-corrected Sea to Sky profile contained similar snowpack features to the original profile (Fig. 9a), despite a dramatic reduction in the amount of precipitation. Similar layers of rounded grains and the same prominent weak layers existed in both profiles. The average similarity value of the two sets of profiles was 0.79, suggesting the profiles were relatively similar in



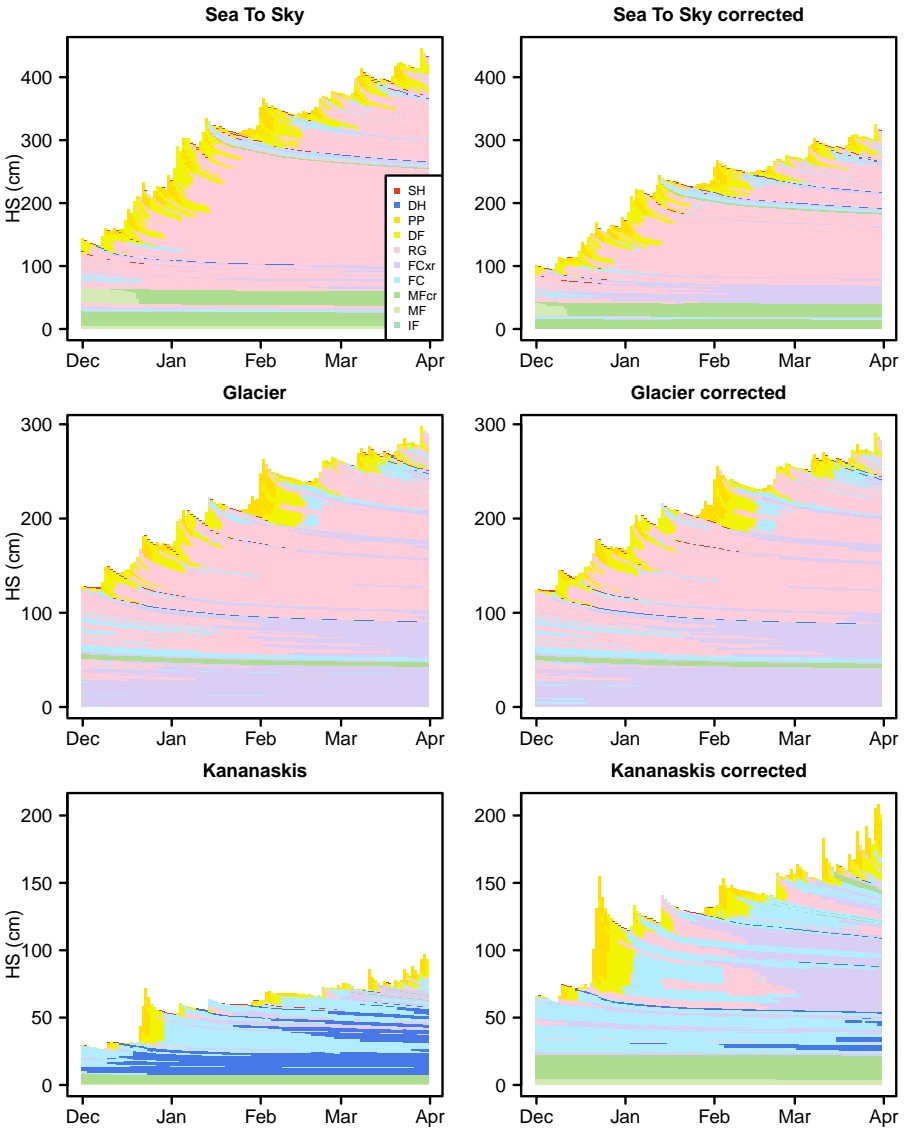

**Figure 9.** Timelines of simulated snowpack structure before and after bias-corrected precipitation inputs at locations in (a) in the Sea to Sky region where modelled snow depths were much larger than observed, (b) the Glacier region where the modelled and observed snow depths closely agreed, and (c) in the Kananaskis region where modelled snow depths were much smaller than observed. Snowpack layers are colour-coded according to grain type (see Fig. 8).




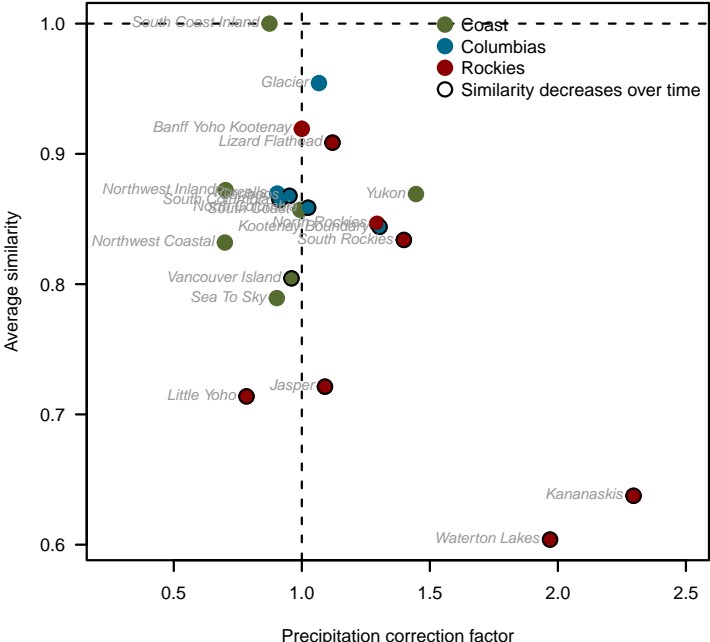

**Figure 10.** Average similarity in snowpack structure between original and bias-corrected snow profile simulations at locations in each forecast region. The similarly is plotted against the magnitude of the bias correction factor applied to the precipitation inputs.

terms of features important to avalanche hazard. The similarly measure did not have a statistically significant change over the course of the season, meaning differences did not compound over time. The bias-corrected profile was effectively a thinner version of the original profile.

The bias-corrected Glacier profile had minimal differences from the original profiles after adjusting the precipitation by a factor of 0.97 (Fig. 9b), with an average similarity value of 0.97 that did not change significantly over time.

The bias-corrected Kananaskis profile, however, had a dramatically different snowpack structure after increasing precipitation by a factor of 2.43 (Fig. 9c). The original profile was primarily composed of a thick depth hoar layer and small amounts of new snow above it, which after adjustment, was stretched into thicker layers of facets, rounding facets, and a few thin well-defined depth hoar layers. The average similarity value of 0.64 suggested the profiles were substantially different in terms of avalanche hazard considerations. Unlike the Sea to Sky and Glacier profiles, the similarity value had a statistically significant decrease

over the course of the season, suggesting the differences compounded over time.

The similarity between the original and bias-corrected profiles are shown for all 21 regions in Fig. 10. The similarity values ranged from 0.60 for the Waterton profiles to 1.00 for the South Coast Inland profiles, with a median value of 0.86. Small bias-corrections generally resulted in profiles that were similar to the original profiles from an avalanche hazard perspective,




as was the case for the Glacier profile. Large adjustments in precipitation in either direction (i.e., increase or decrease) resulted
in more substantial changes. The impacts of bias-corrections were most dramatic for profiles in the Rockies, as the four
locations with the lowest similarity values (all less than 0.75) were all in the Rockies. Profiles in the Coast and Columbias had
similarity values greater than 0.8 (except for the Sea to Sky profile), suggesting bias-corrections had relatively smaller impacts
on simulated avalanche conditions. When comparing two locations with relatively small bias-corrections, a 0.97 correction
in Glacier resulted in a similarity of 0.95 compared to a similar correction of 0.98 in Banff Yoho Kootenay that resulted in
a similarity of 0.92. In this case, the small adjustment in precipitation in Banff Yoho Kootenay resulted in more substantial
changes to the snowpack structure. The similarity value became statistically lower over the course of the season in 1 of 7
Coast profiles, 4 of 7 Columbias profiles, and 6 of 8 Rockies profiles, suggesting the impact of bias-corrections become more
compounded over time in colder continental climates.

## 5   Discussion

### 5.1   Implications for weather forecasting

Lundquist et al. (2020) argue the increasing skill in modelling mountain precipitation relative to traditional observations
networks warrants new multi-disciplinary methods for evaluating models. Avalanche safety operations spend considerable
effort collecting weather observations at high elevations, which could be a valuable addition to verifying weather forecasts.
Avalanche weather observations are rarely included in meteorological data assimilation or analysis products (Roy et al.,
2018; Snauffer et al., 2018), as they rarely meet meteorological standards (e.g., World Meteorological Organization). One
challenge is that most meteorological products measure precipitation in millimetres of water equivalent, but the coverage and
quality of precipitation gauges in mountainous terrain is relatively much poorer than snow depth observations. Our analysis of
available datasets reveals notable patterns in the performance of the HRPDS weather model in western Canada, including the
overprediction of precipitation on the windward side of the Coast range (Fig. 5). This specific bias likely originated from the
NWP model's handling of hydrometeor drift and spillover in complex terrain (Mo et al., 2019), which is a difficult to calibrate
parameterization. Additional analyses (not shown) using snow water equivalent and precipitation measurements from the same
observation networks did not provide as clear results. Therefore, it would be worthwhile to incorporate more snow observations
into weather products, as done with the SNODAS product produced in the contiguous United States (Barrett, 2003).

This study highlights several shortcomings of mountain observation networks across western Canada, as suggested by
Lundquist et al. (2020). For example, many regions had sparse and intermittent observations that result in large uncertainties
when interpolating snow depths (Fig. 3a). Model verification metrics often performed poorer in areas with greater observation
uncertainty (Fig. 7), suggesting NWP models may not actually perform as poorly as verification metrics suggest and that
observations should not be treated as absolute ground truth. The regression models highlighted that in many regions the expert
assessed avalanche conditions were explained better with modelled snow depth changes than with observed snow depth changes
(Table 1). In particular, avalanche conditions in the heavy snowfall regions in the Coast range were better explained by modelled
snowfall than observations (Fig. 4b-c). While this could be partly explained by the fact avalanche forecasters put more weight





on NWP model forecasts in their assessments in these regions, it could also be explained by shortcomings of the observation networks in these harsh coastal environments. Except for small data-rich regions like Glacier, regression models with modelled snowfall were relatively similar in performance to models with observed snowfall. While these regression models were limited
by human assessment errors and by over-simplifying the factors influencing avalanche hazard, they provide a simple example of how model information is comparable to observation information.

## 5.2   Implications for snowpack modelling

Given precipitation has been shown to be a primary source of uncertainty in snowpack simulations (Raleigh et al., 2015; Richter et al., 2020), meaningful methods to verify and correct erroneous precipitation inputs could dramatically improve the quality
of snowpack models. However, this study highlights large uncertainties in many observation networks that warrant careful approaches when evaluating snowpack models.

The comparisons of modelled and observed snow depths across western Canada presented in this study suggest a large-scale precipitation bias existed with too much precipitation in Coast range and too little precipitation in the Rockies (Fig. 4 and Fig. 5). Correcting these biases with a constant correction factor, as done in this study, could be an appropriate method in areas with
high quality observations. For example, many locations in the southern Coast range had a strong correlation between modelled and observed HN (Fig. 7) as well as relatively high confidence in the observations (Fig. 3), suggesting precipitation errors were likely spread evenly across all precipitation events. This would be a good candidate for applying a constant bias-correction factor to snowpack simulations. However, a constant bias correction may be less appropriate in the northern Coastal range where observed HN had weaker correlations with modelled depths and the observations had greater uncertainty.

Bias corrections would only be useful when these biases persist in an NWP model over several seasons, but ideally NWP models continually improve their precipitation forecasts by increasing grid resolution and improving their data assimilation, physics, and parameterizations. A more robust method would be to correct precipitation errors at finer time scales with a data assimilation routine like the one presented by Winstral et al. (2018). This dynamic bias correction method would be beneficial in areas with sufficient observation networks to capture short term changes in snow depth, but as found in this study, this could
be challenging in large regions where sparse and inconsistent observations do not provide a clear picture of short-term snow depth changes.

While detailed assimilation was outside the scope of this study, adjusting precipitation inputs by constant factors illustrated some key impacts of precipitation errors from an avalanche hazard perspective. The greatest change in snowpack structure was observed in the continental Rocky Mountain range (Fig. 10). In cold climates, changes in snow depths had a greater impact
on temperature gradients in the snowpack, and as a result, the formation of weak faceted layers. Increasing precipitation could result in substantially fewer faceted layers and a less hazardous snowpack structure, while decreasing precipitation could result in substantially more faceted layers and a more dangerous structure. Locations in the Coast range, on the other hand, exhibited fewer differences in their snowpack structure after bias corrections. Changing snow depths in maritime climates has less impact on the temperature gradients, and as a result the bias-corrected snowpack resembled a stretched or compressed version of the
original profiles, usually containing the same weak layers and crusts.



### 5.3 Implications for avalanche forecasting

This study found relatively strong agreement between modelled and observed snow depths in many situations across western Canada. In cases with strong agreement, the simulated snowpack structure can be interpreted with a higher degree of confidence. In situations where the modelled and observed snow depths differ, the simulated snowpack structure must be evaluated more critically. Considerations should include the representativeness of local observations, whether the model has a known precipitation bias in that region, and the sensitivity of the snowpack structure to snow depth errors in that climate.

Assessing the quality of snow depth observations identified regions where forecasters deal with data sparsity and uncertain observations (Fig. 3). Snowpack models could be particularly valuable in these regions, but to interpret the models it would help to collected targeted snow depth information from automated weather stations, field observers, crowd sourcing platforms (e.g., Mountain Information Network, Community Snow Observations), and satellite-derived snow cover products.

This study focused on regional-scale patterns to provide a general understanding of model performance; however, snowpack models also have potential to simulate finer scale spatial patterns. While such patterns are highly relevant to avalanche forecasters, they become even more difficult to verify with sparse observations. Therefore, we suggest snowpack models forced with NWP model output should first be understood at coarse regional scales. Exploring applications at finer scales could be considered in areas with high quality observations and strong agreement between modelled and observed snow depths. Aggregating observations across a common reference treeline elevation provided a consistent approach for regional-scale comparisons, however, more precise aggregations such as sub-grid parametrization of sky view factor (Helbig and van Herwijnen, 2017) would be required for verifying snow depth at smaller scales. The comparisons presented in this study were also point-wise, where modelled and interpolated observations were compared at the same location, but spatial verification techniques such as neighbourhood and feature-oriented approaches could also be meaningful to understand regional-scale patterns relevant to avalanche forecasters (Gilleland et al., 2009).

### 6 Conclusions

Forcing snowpack models with output from high-resolution NWP models is a promising method to support avalanche forecasting. However, there is limited observation data available for verification in many situations. Furthermore, the skill in modelling mountain precipitation continues to improve to the point where it outperforms observation networks in many contexts. Aggregating snow depth observations from networks such as automated weather stations operated by government services and manual observations from professional avalanche observers can provide insights into model performance at coarse spatial and temporal scales.

Applying this approach over a winter season across the diverse mountain climates of western Canada provided several insights about using snowpack models for avalanche forecasting. First, the quality of weather observations should be considered when verifying NWP and snowpack models. The number and consistency of observations in some areas resulted in relatively large uncertainties when interpolating snow depth, and in roughly half of the forecast regions, the avalanche conditions assessed by expert forecasters were explained with modelled snow depth changes better than with observed snow depth changes.

Despite limitations in the observations, the comparison of modelled and observed snow depths strongly suggested a positive
precipitation bias existed in the Coast range and a negative bias existed in many parts of the Rockies. The agreement in snow
depth was strongest in the transitional Columbia Mountains, which also had the highest density of quality observations. The
potential impacts of snow depth errors were illustrated by comparing the snow profiles produced with NWP models output
with profiles produced with bias-corrected precipitation inputs. Precipitation bias-corrections had the most dramatic effect in
cold continental climates where the snow depths heavily influenced the degree of faceting. In warmer maritime climates, bias
corrections resulted in stretched or compressed profiles with similar snowpack structure.

These results highlight how meteorologists and NWP model developers could benefit from the observation networks of
avalanche safety operations as well as their subjective knowledge of mountain weather. Snow depth observations could be
a particularly valuable data stream, and NWP models could benefit from additional research into methods for assimilating
snow observations. Improving snowfall forecasts from NWP models would have direct benefits to avalanche forecasting and
snowpack modelling. From an avalanche forecasting perspective, this study highlights how limitations in observation networks
pose a challenge to verifying snowpack models, and how these limitations need to be carefully considered when interpreting
simulated profiles for avalanche forecasting. This provides a starting point for future research into how operational snowpack
models could perform real-time verification and assimilation with available snow observations.

. Code and data are publicly available on the Open Science Framework at https://osf.io/a5pek (Horton and Haegeli, 2022).

## Appendix A:  Original and bias-corrected profiles for all regions

. Both authors conceptualized the research with SH leading the analysis and writing and PH providing supervision.

. The authors declare that they have no conflict of interest.

. Thanks to Avalanche Canada, the Canadian Avalanche Association, and InfoEx subscribers for sharing datasets. Patrick Mair, Florian
Herla, and Stan Nowak provided valuable input on data analysis and visualization methods.





**Figure A1.** Timelines of simulated snowpack structure before and after precipitation bias-correction. Two profile timelines are shown for each forecast region, with the original profiles to the left and the bias-corrected profiles to the right with the bias correction factor ($k$) shown in the title. Snowpack layers are colour-coded according to grain type (see Fig. 8).



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
