# Peer review of "Using snow depth observations to provide insight into the quality of snowpack simulations for regional-scale avalanche forecasting"

_EGUsphere, 2022_

## Author Comment (AC1)

**Author response to RC1 and RC2 for "Using snow depth observation to provide insight into the quality of regional-scale snowpack simulations for avalanche forecasting"**

Simon Horton

June 29, 2022

Dear editor and reviewers,

Thank you for the constructive comments. We would like to submit a revised manuscript to address your feedback with the following main changes:

1. Change our description of the observation dataset throughout manuscript to emphasize our intent is to upscale point observations to a regional-scale and evaluate snowpack models at that scale because it is relevant to avalanche forecasters. We updated our motivation and description of this approach and changed our terminology to describe this data as "upscaled observation data" rather than "observations". This allows us to be more specific when discussing the types of uncertainties and the limitations of using point observations for regional-scale assessments.

2. Improve our explanation of how modelled snow heights are impacted by more than just precipitation errors. With better framing and acknowledgement of these difference, we argue our methods are still appropriate to address our research question. We replace terminology such as "bias-corrected profiles" with "precipitation-adjusted profiles" to better reflect that we are not actually assimilating observations, and have removed some of the discussion about assimilation.

3. Extract a new dataset of modelled snow depths using grid points that better represent regional-scale conditions. Rather than choosing grid points with the closest match to treeline elevation in each 10 km grid cell, we select the grid point with the median accumulated precipitation over the season. This method smooths some of the random noise from the previous grid point selection method. We prefer this approach to averaging values from multiple grid points to avoid inconsistencies that arise from averaging the multiple weather variables.

4. Replace our analysis of the new snow height variable ($HN$) with an analysis of the daily change in snow depth ($\Delta HS$). This had little impact on the results, but provides a simpler and transparent method that can be applied to different data sources in a consistent way.

5. Update all analysis, results, and figures with these new datasets.

6. Improved the quality of the figures.

The main impact of these revisions is that our updated description of observation data and precipitation adjusted profiles better reflect the objective of our study. Updating the analysis with new model and observation data has produced slightly different results, but the main discussion points and conclusions remain the same. Please see below for our point-wise responses to the reviewer's comments (comments in italics and responses in bold).

Best,
Simon Horton

**Response to RC1 (Anonymous reviewer)**

*The manuscript entitled 'Using snow depth observation to provide insight into the quality of regional-scale snowpack simulations for avalanche forecasting' by authors Horton and Haegeli discusses the potential of snow cover models while forced with forecasted data to provide additional information on the snow cover on the regional scale especially for the regions where observations are sparse. In particular, this study focuses on assessing or quantifying the quality of such simulations for different regions with different snow climates across Western Canada with the overall goal to identify regions with high or low confidence in these model simulations. The paper is well written and structured and provides valuable in-sight into the benefits as well as shortcomings of such model chains for avalanche forecasting and other applications.*

*Specific comments*

*As I understand from the manuscript forecasted precipitation amounts of a single grid point were used to force the snowpack model. Although taking the closest grid point with the smallest vertical difference to the location of interest is meaningful, it is also common practice in verification of forecasted precipitation amounts to use an average of at least 9, i.e. closest grid cell plus 8 surrounding cells. Selecting a single grid point might represent the tree-line elevation, but might not represent orographic effects and the grid point might get less or more precipitation depending on the prevailing wind. E.g. in line 369 the authors state that HRPS does overpredict precipitation on the windward side of the Coast range. Could the authors comment on the effect of using a single grid point in particular for precipitations amounts from a single grid point instead of an average of multiple points on their results? To be more precise. How do results change if more than one grid point is used?*

We agree the original method of choosing grid points with the smallest vertical difference can create undesired randomness. High-resolution NWP models can resolve some local orographic effects within 10x10 km grid cells that results in substantial variations in forecast precipitation. To better represent treeline conditions at regional scales, we updated our method to choose a different set of grid points. Instead of the selection criteria being closest vertical match within a cell, we select points based on their total accumulated precipitation over the 2020-21 winter season. We consider all grid points that are within 100 vertical meters of treeline in each grid cell and then choose the point with the median accumulated precipitation. This prevents selecting points that are relatively too dry or too wet within a grid cell, while the 10 km cell resolution allows resolving the major differences between wet and dry areas within each forecast region.

We have experimented with several grid point sampling methods in our operational snowpack modelling system, including averaging multiple grid points. We find this can have undesired effects. When averaging all the input variables for SNOWPACK there can be internal inconsistencies such as non-physical combinations of precipitation and humidity, temperature and radiation, etc. Choosing a single grid point preserves the internal consistency of the meteorological forcings.

Repeating the analysis with a different grid points has changed the results in subtle ways. The maps and statistical metrics are different at finer scales, however when the results are aggregated to the scale of forecast regions, the main results and their interpretations are the same.

*Please indicate how SNOWPACK was forced. Incoming short and long wave radiation? Surface Temperature? Air temperature (2m diagnostic air temperature or first atmospheric level? Although, as also stated by the authors, simulations are most sensitive to precipitations amounts the other meteorological parameters have also an impact on the simulations. Please comment.*

We now list the six meteorological forcings used for SNOWPACK in Data Sect. 2.2. We also better explain throughout the manuscript how the modelled snow depths we evaluate are driven by more than just precipitation inputs. This includes some specific discussion in the Introduction and rewording references to our sensitivity study as "precipitation adjustments" instead of "bias-corrections", which reflects the fact we do not actually measure precipitation or calculate precipitation biases.

*Using a correcting factor k (Equation 6) for precipitation amount solely based on observed and modeled snow depth seems a little dangerous and maybe not very meaningful, because different snow heights might not stem from the inadequate modelling of precipitation amounts alone but rather from different new snow densities due to different forecasted air temperatures and windspeeds. Please comment or elaborate a little further around*

*Lines 318-320.*

**We better acknowledge the difference between snow depth errors and precipitation errors throughout the manuscript, including our Methods section where we have made modifications to better highlight the limitations of our approach. We further justify this method as a simple way to illustrate impacts in different snow climates.**

*Technical comments*

*No technical comments. As stated above the manuscript is well written.*

**Response to RC2 (Matthieu Lafaysse)**

*General comments*

*In this paper, Simon Horton and Pascal Haegeli address the necessary but challenging task of evaluation of a snow modelling system in support of avalanche hazard forecasting. To address this question, they transform local snow depth observations in a regional scale assessment of snow depth at treeline before the comparisons with numerical simulations. They also compare the predictability of a simple statistical model of avalanche hazard using predictors from either the model or the observations. They finally illustrate the impact of precipitation errors on the simulated stratigraphies. The paper is well written and well structured, with interesting results supporting the discussion. Some methodological choices are unusual, which is of course interesting and probably the main added value of the paper for the community, but these choices would have sometimes required, to my mind, a better justification.*

*Mainly, the inconsistent spatial scale between snow observations and model simulations is a very well-known problem in spatialized snow modelling evaluation. A strong choice of the methodology of this paper is to adapt observations towards the modelling geometry as described in Section 3.1 rather than adapting model output to observation locations or more simply filtering data with too much spatial discrepancies. Although all methods have advantages and disadvantages, the approach used here is not common compared to previous literature which often evaluate models directly with raw observations without any interpolation or spatial aggregation of observations. There are probably good reasons for using such a specific approach here (specificities of non-conventional observations? scale of interest for avalanche forecasters?), but I would have expected a better justification and discussion of this choice in the paper. Why interpolating observations rather than model outputs? How this can affect the conclusions? Does it not amplify our perception of observation uncertainty rather than model uncertainty? Indeed, all the correction factors in Section 3.1 are very likely to add a significant level of uncertainty rather than considering a snow depth observation as it is, i.e. only representative of the point where it's done. Perhaps, it would also help to introduce this challenge of spatial scale in a more explicit way in the introduction. My feeling is that a significant part of what the authors identify here as uncertainty of "observations" would have been considered in common model evaluations as "unresolved spatial variability" of the simulations. This can be obviously debated, but I think the introduction of the challenge and the discussion of the pros and cons of the methodology compared to existing literature could be improved in the paper.*

**Thank you for the fair critique of our methods. We agree our approach is inherently different from classic model evaluations, and we have taken your advice to better emphasize and justify this throughout the manuscript. Rather than referring to our dataset of interpolated point observations as "observations" we now call these "upscaled observations". This better reflects the idea that avalanche forecasters need to evaluate snowpack models at regional scales. It also makes it clearer when we discuss limitations of the observation network that we mean they have limitations for representing regional-scale conditions. We bring out the issue of matching spatial scales more explicitly in our Introduction and Methods sections, so it is clearer how our results should be interpreted. We are also more specific that our arguments are for the context of "regional-scale" applications, especially data-sparse regions.**

*Detailed comments*

*L26 Indeed these references assess the ability of Crocus to simulate optical reference but it also worth mentioning that optical satellite observations are also often reduced to a simple Snow Cover Fraction, which is a common evaluation variable in snow modelling (many available references in the snow hydrology community).*

This paragraph was intended to summarize Morin et al. (2020) Sect 6.4 (Information quality: Integrity) to show there are a variety of ways to verify snowpack models, but they all deal with issues of uncertainties and spatial representativeness. We updated to first sentence to better articulate this. We do not try to provide a comprehensive list and description of these methods and going into specific details about target variables (e.g., snow cover fraction, snowpack tests) would distract from our main point that these methods are difficult to apply in real-time.

*L30-32 A number of the stations used in the mentioned references also provide real-time observations and are used in real-time monitoring of snow modelling systems.*

True, we added sentences to discuss the more recent studies that use networks that are suitable for real-time monitoring such as Quéno et al. (2016), Vionnet et al. (2019), Winstral et al. (2018), and Cluzet et al. (2022). Most notably, Quéno et al. (2016) was an important influence on this study and should have been cited in the original manuscript.

*L38-42 Although I acknowledge that observation uncertainties and spatial representativeness must be accounted for in model evaluations, at the current state of the art of snow modelling, I honestly think it is more than optimistic to consider than snow simulations can outperform the accuracy of snow observations at the local scale. The last sentence of the paragraph is definitely very far from the perception of snow modelling by French avalanche forecasters ! I would recommend to be more specific on the contexts, and especially to limit the spatial scale for which this statement applies.*

We are more specific about the context where we think model and observation skill is comparable. We appreciate the European perspective, where observations are likely much more reliable than many of the remote data-sparse situations we are used to in Canada. We are more specific about representing "region-scale" conditions to acknowledge issues of mismatched spatial scales.

*L43-48 It is true than precipitation forcing is always found as the main source of uncertainty of snow modelling, but other uncertainties can not be ignored. Especially snow depth simulations are also known to be especially sensitive to the accuracy of longwave incident radiations (Raleigh et al, 2015; Sauter and Obleintner, 2015; Quéno et al. 2020). They can also be affected by very uncertain parameterizations of new snow density (Helfricht et al., 2018). Therefore, it should be more clear than the evaluations performed in this study assess the ability of the whole system to simulate snow depth (including all forcing errors and snow modelling errors, but not reduced to precipitation errors).*

We agree that our manuscript did not adequately acknowledge the difference between precipitation input and modelled snow depth. We updated this paragraph and other sections to better communicate that our approach is evaluating the potential impacts of precipitation errors on the interpretation of avalanche conditions in different regions, rather than suggest snow depth biases should be used to correct precipitation errors.

*L59-60 The limitation of data to the end of March has a strong impact on the scope of the study, which should be better emphasized. Indeed, it is rather clear from Figure 6 that this paper only focuses on the snow accumulation period and that the melting period is excluded from the analysis.*

We add a statement about the specific scope of this study being the snow accumulation period when dry snow avalanches are the focus for forecasters. To evaluate snowpack models during the melting period we would likely have modified our approach, but at least in our context, operational forecasters are more interested in the accumulation period.

*L83-90 I understand the choice to sample simulation points to reduce numerical costs, but indeed in that case as mentioned by the previous reviewer, it is questionable to select only the closest point rather than smoothing NWP output among different points of the 10 km grid cell, especially in the context where these simulations are going to be compared to spatially smoothed observations.*

We have changed our sampling method to get smoother results by choosing grid points with median accumulate precipitation. See our response to the previous reviewer for details.

*L109 I don't understand the choice of summing hourly variations of HS to obtain 24h height of new snow. Indeed, the definition of height of new snow in the International Classifications does include the impact of settlement*

*of new snow, melting, or any other process modifying the snow depth during the 24 hours, as the reference measurement of this variable is a snow board where all these processes occur. When using HS to derive HN, the problem of settlement below the new snow also exists, but it is not solved by the sum of hourly values. Can you better justify this choice or maybe redefine the evaluated variable if too different with the standard concept of height of new snow? Note that daily snow depth variations is also a useful evaluated variable (Quéno et al., 2016 ; Vionnet et al., 2019).*

For simplicity and transparency we changed our new snow variable to daily snow depth change $\Delta HS$, as done in the references above. This resulted in only minor changes to our results. Our previous method is not standard, but rather based on an experimental method that has provided practical estimates of new snow from the AWS stations used by Avalanche Canada.

An interesting impact of switching to $\Delta HS$ for both model and observation data is that more of the regression models perform better with observation data than with model data, and that overall the models perform poorer than with the previous more complex estimates of new snow (i.e., $HN$ from the model, hourly sums from AWS, and manual storm board measurements). This suggests there is added value in forecasters using the snowpack models simulated new snow depth over a simple HS change.

We think a detailed comparison of methods to derive new snow amounts from different sources of data is outside the scope of our study, and so using $\Delta HS$ is a simple and intuitive choice to illustrate how well our data sets represent regional-scale avalanche conditions. It could be interesting to use a similar regression model approach to evaluate different methods of deriving new snow amounts.

*L181 Does this variance really represent the uncertainty of observations or does it simply represent the small scale spatial variability of snow depth which is known to be very high? Maybe another way to consider the question is should we consider your regional assessment of snow depth at treeline as an observation considering the complex and uncertain protocol necessary for this assessment?*

See our response to the general comment above. In this specific example, we argue the interpolation variance represents the uncertainty in observations representing regional-scale conditions due to issues of observation density and the influence of local-scale processes. We no longer call this an uncertainty in the observations, but rather uncertainty in the "upscaled observations", which we think better reflects the context and source of this uncertainty.

*L228-229 The bias correction method used here is probably sufficient to investigate the sensitivity of snow profiles to precipitation errors. However, I recommend to emphasize here that (1) the assumption behind this method is that snow depth errors are entirely explained by precipitation errors, which is a very strong simplification (see my comment about L43-48), and (2) that this correction method is not the state-of-the-art way to assimilate snow depth observations in a snow model (Largeron et al. 2020, Cluzet et al., 2022, I give references from my team but of course feel free to use other ones as many teams work on that topic).*

We better acknowledge the difference between snow depth errors and precipitation errors throughout the manuscript, including our Methods section where we have made modifications to better highlight the limitations of our approach, while justifying it is a simple way to illustrate the impacts of these errors in different snow climates. Also, it was promising to see the recent and complimentary work on this topic done by Cluzet et al. (2022). We updated our citations and look forward to sharing our experiences in applying these methods in operational systems.

*L246 Again, I am wondering if the word uncertainty is appropriate as it might be associated with measurement errors when it is actually mainly refers to subgrid spatial variability.*

In this paragraph we are more specific that there is uncertainty when upscaling available observations to a regional-scale.

*L272-274 Does it really make sense to compute a spatial correlation between simulations and interpolated observations when the number of real observations for some subregions is only 1 or 2 stations? I think it means this metric just reflects the ability of the interpolation method itself to explain the simulated variability of snow depth but it is poorly related to the ability of simulations to explain an observed spatial variability. The same question applies for regions where only a very low number of simulated grid points (¿=3) are considered.*

We agree the correlations are less meaningful in cases with fewer location to compare, but are

still informative because in these smaller regions we are still interested in learning whether the model and available observations can resolve the relatively deeper and shallower areas. For example, regions with few locations in the Rockies generally had poorer correlations than similar size regions in the Coast Range, suggesting the model was likely better at resolving regional-scale gradients on the coast where there is a stronger windward/leeward precipitation gradient. Poor correlations in small regions could also helps forecaster identify potential regions with poor observation density. We add a few sentences in the discussion to provide our interpretation of the correlations in small regions.

*L287 Unfortunately, it is not possible to identify in the maps the position of this transect as (1) the transect is not materialized in any map, and (2) the maps do not provide the geographical coordinates. Could you improve this?*

We added latitude and longitude grid lines to all of our maps and identified the specific latitudes of this transect on the map in Fig. 4a.

*Figure 8 The legend for grain types colors is really tiny. Could you add a common and larger legend bar below the Figure?*

Yes, we added a legend bar below the snow profile plots to make it easier to see the grain types colours.

*L317 I agree it helps to have a correct snow depth, but this is not sufficient to guarantee an appropriate stratigraphy, and this should be remind for readers unfamiliar with detailed snow modelling.*

A comment was added to remind readers that snow depth agreement does not guarantee correct stratigraphy.

*L364 Note that surface precipitation from rain gauges are almost never assimilated in the assimilation cycles of NWP systems, even for rainfall in low lands, so this is not specific to snow observations from avalanche networks. I generally agree with this discussion, but maybe you could limit this comment to the development of analysis products and evaluation of NWP, but remove the reference to data assimilation in NWP.*

We removed a substantial amount of discussion about assimilation because, as pointed out by the reviewers, adjusted precipitation inputs based on snow depth differences should not be treated as proper assimilation, and instead frame this analysis as an exploration of the impact regional-scale biases have on simulated avalanche conditions. Our discussion is now focused on the added value of snow depth observations for regional-scale forecasting, especially in data sparse areas.

*Also, it could be mentioned that in some countries (France), the density of snow observation networks and of precipitation observations are unfortunately correlated, which limits the potential added value of incorporating snow observations in analyses products (because they are available only where the precipitation network is already sufficiently dense). This is especially emphasized in Cluzet et al., 2022. This is not the case in Switzerland, where a very dense snow observation network almost everywhere has on the contrary a strong positive impact on precipitation analyses.*

We add a comment to explain where the greatest added value of HS observations would be.

*L374-378 This discussion raises again the same ambiguity as mentioned before. The point is that observations should definitely be preferred as ground truth compared to numerical simulations, as long as they are considered at their appropriate spatial scale (local and not regional). The uncertainty of interpolation observation products may indeed be higher than uncertainty of numerical models, in their ability to estimate regional snow depth. But I really think it is important to not mix up observations and interpolation of observations, and not mix up local scale and regional scale. Therefore, too general sentences as "observations should not be treated as absolute ground truth" are to my mind inappropriate.*

This paragraph was carefully re-written to better describe our uncertainties interpreting point observations at regional scales.

*L394-395 I think that the correction method used in this study was fine to illustrate the impact of these errors on snow stratigraphies. However, even with high quality observations, I don't think that this method should be*

*recommended for an operational system as more advanced data assimilation techniques exist to avoid the strong assumptions of (1) temporally and spatially homogeneous precipitation errors and (2) seeing precipitation errors as the unique source of snow modelling errors.*

**We abbreviated this discussion since evaluating assimilation methods was outside the scope of our study. We now cite relevant literature on state-of-the-art assimilation and make arguments about the need for this based on our findings.**

*L445 quality or density?*

**Changed to "quality and density".**

*Despite these comments, the necessity of this paper is obvious in the context of the development of a new snow modelling system for Western Canada, and I like the idea to not only consider the classical metrics to compare simulations and observations but also to compare their ability to predict avalanche hazard.*

**Thank you. This study was inspired from our experience using snowpack models for operational forecasting. We noticed simulated snowpack conditions could be particularly valuable for regional-scale hazard assessment in areas with poor weather station coverage, however there were situations where the snow depths differed substantially from our knowledge of the region and we were uncertain how to interpret the profiles. We also know that forecasters' knowledge of regional-scale conditions is informed by much more information than the AWS networks, and it would be helpful to have ways to apply this knowledge to interpret and potentially correct NWP and snow cover models.**

*References*

- *Cluzet, B., Lafaysse, M., Deschamps-Berger, C., Vernay, M., and Dumont, M. : Propagating information from snow observations with CrocO ensemble data assimilation system : a 10-years case study over a snow depth observation network, The Cryosphere, 16, 1281-1298, https://doi.org/10.5194/tc-16-1281-2022*

- *Helfricht, K., Hartl, L., Koch, R., Marty, C., and Olefs, M.: Obtaining sub-daily new snow density from automated measurements in high mountain regions, Hydrol. Earth Syst. Sci., 22, 2655-2668, https://doi.org/10.5194/hess-22-2655-2018, 2018.*

- *Largeron C., Dumont M., Morin S., Boone A., Lafaysse, M., Metref S., Cosme E., Jonas T., Winstral A. and Margulis S.A. (2020) Toward Snow Cover Estimation in Mountainous Areas Using Modern Data Assimilation Methods : A Review. Front. Earth Sci. 8:325. doi : 10.3389/feart.2020.00325*

- *Quéno, L., Vionnet, V., Dombrowski-Etchevers, I., Lafaysse, M., Dumont, M., and Karbou, F. : Snow-pack modelling in the Pyrenees driven by kilometric-resolution meteorological forecasts, The Cryosphere, 10, 1571-1589, doi:10.5194/tc-10-1571-2016)*

- *Quéno, L., Karbou, F., Vionnet, V., and Dombrowski-Etchevers, I.: Satellite-derived products of solar and longwave irradiances used for snowpack modelling in mountainous terrain, Hydrol. Earth Syst. Sci., 24, 2083-2104, https://doi.org/10.5194/hess-24-2083-2020, 2020.*

- *Raleigh, M. S., Lundquist, J. D., and Clark, M. P.: Exploring the impact of forcing error characteristics on physically based snow simulations within a global sensitivity analysis framework, Hydrol. Earth Syst. Sci., 19, 3153-3179, https://doi.org/10.5194/hess-19-3153-2015, 2015.*

- *Sauter, T. and Obleitner, F. : Assessing the uncertainty of glacier mass-balance simulations in the European Arctic based on variance decomposition, Geosci. Model Dev., 8, 3911-3928, doi :10.5194/gmd-8-3911-2015*

- *Vionnet, V., Six, D., Auger, L., Dumont, M., Lafaysse, M., Quéno, L., Réveillet, M., Dombrowski-Etchevers I., Thibert, E. and Vincent, C. : Sub-kilometer precipitation datasets for snowpack and glacier modeling in alpine terrain, Front. Earth Sci., 7, 182, https://doi.org/10.3389/feart.2019.00182*

**We added citations to Cluzet et al. (2022), Helfricht et al. (2018), Largeron et al. (2020), Quéno et al. (2016), and Vionnet et al. (2019). We were not familiar with the first three, while the last two should have been cited in the original manuscript since their heavily influenced our study.**

---

## Author Response (AR2)

Dear Cryosphere,

Thank you for accepting our revised manuscript. Our uploaded files contain all the technical corrections suggested by the handling editor and reviewers.

Best,

Simon O. Vorton